# ITGA2 Gene Polymorphism Is Associated with Type 2 Diabetes Mellitus in the Kazakhstan Population

**DOI:** 10.3390/medicina58101416

**Published:** 2022-10-09

**Authors:** Aizhan Magazova, Yeldar Ashirbekov, Arman Abaildayev, Kantemir Satken, Altynay Balmukhanova, Zhanay Akanov, Nurlan Jainakbayev, Aigul Balmukhanova, Kamalidin Sharipov

**Affiliations:** 1Aitkhozhin Institute of Molecular Biology and Biochemistry, Almaty 050012, Kazakhstan; 2Department of Biochemistry, Asfendiyarov Kazakh National Medical University, Almaty 050012, Kazakhstan; 3Department of Health Policy and Organization, Al-Farabi Kazakh National University, Almaty 050040, Kazakhstan; 4City Center of Diabetes AAA Medical Clinic, Almaty 050040, Kazakhstan; 5Department of Public Health, Kazakh-Russian Medical University, Almaty 050004, Kazakhstan; 6International Medical School, Caspian University, Almaty 050000, Kazakhstan

**Keywords:** type 2 diabetes mellitus, diabetic retinopathy, α2β1 integrin receptor, BglII polymorphism in ITGA2 gene

## Abstract

*Background and Objectives:* Nowadays, every tenth adult in the world suffers from diabetes mellitus (DM). Diabetic retinopathy (DR) is the most common microvascular complication of type 2 DM (T2DM) and a leading cause of acquired blindness in middle-aged individuals in many countries. Previous studies have identified associations of several gene polymorphisms with susceptibility to microvascular complications of DM in various worldwide populations. In our study, we aimed to test the hypothesis of the associations of single nucleotide polymorphisms (SNP) of the VEGF (−2549I/D), RAGE (−429T/C and −374T/A), TCF7L2 (rs7903146), and ITGA2 (BglII) genes with a predisposition to DR among T2DM patients in the Kazakhstan population. *Materials and Methods:* We conducted a case–control study comparing the genotype distribution and allele frequencies between groups of DR patients (*N* = 94), diabetic patients without DR (*N* = 94), and healthy controls (*N* = 51). Genotypes were identified using the PCR-RFLP method. *Results:* In all cases, the genotype distribution corresponded to the Hardy–Weinberg equilibrium. The groups of diabetic patients with and without DR did not significantly differ in the genotype distribution of the SNPs studied. Differences between both groups of diabetic patients and healthy controls in four out of five SNPs were also not significant. At the same time, both groups of diabetic patients differed significantly from healthy controls in genotype distribution (*p* = 0.042 and 0.005, respectively) and allele frequencies (*p* = 0.021 and 0.002, respectively) of the BglII polymorphism in the ITGA2 gene. After adjusting for multiple comparisons, the differences between the group of diabetic patients without DR and the control group remained significant (*p_Bonf_* = 0.027 for genotypes and *p_Bonf_* = 0.009 for alleles). The BglII− allele was associated with diabetes: OR = 1.81 [1.09–2.99] for DR patients, and OR = 2.24 [1.34–3.75] for diabetic patients without DR. The association was also observed in the subset of Kazakhs. *Conclusions:* This study shows that the BglII polymorphism in the ITGA2 gene can be associated with T2DM but not with DR. According to our data, the risk allele for diabetes is the wild BglII− allele, and not the minor BglII+, which is considered as risky for DR.

## 1. Introduction

The WHO has defined the situation of diabetes mellitus (DM) as an epidemic of a non-infectious nature due to its wide prevalence, early disability of patients of working age, and high mortality. According to the International Diabetes Federation, today in the world, one in ten of the adult population (aged 20 to 79 years) suffers from diabetes, which is around 537 million people, and this number is estimated to increase to 783 million by 2045 [1]. DM patients suffer from many complications of the disease, including macrovascular (stroke, coronary artery disease, and peripheral arterial disease) and microvascular (retinopathy, neuropathy, and nephropathy) pathologies [2]. Diabetic retinopathy (DR) is the most common microvascular complication of diabetes (every third type 2 DM (T2DM) patient has DR) as well as the leading cause of acquired blindness in middle-aged people in many countries, and it also tends to increase in incidence [3].

DR is characterized by leaky retinal vasculature, retinal ischemia, angiogenesis, and retinal inflammation. These pathologies are clinically manifested as cotton-wool spots, exudates, small tortuous veins, aneurysms, and areas of hemorrhage, which lead to decreased visual acuity, loss of color sensitivity, and impaired night vision [4]. Retinal inflammation contributes to increased vascular permeability and blood–brain barrier disruption, resulting in diabetic macular edema (DME), which is the first complication of DR, leading to reduced central vision [5,6]. The second complication develops due to retinal ischemia causing the growth of new small pathological blood vessels in the central part of the posterior segment that attach to the surface of the vitreous and are prone to rupture and result in retinal detachment, mostly known as proliferative DR (PDR) [6,7].

Clinical data have shown that some diabetic patients, despite the long duration of their disease (25 years or more), have no signs of DR or have a manifestation of minimal non-proliferative DR (NPDR) [8]. These data, along with the observed family correlation of DR, indicate the existence of a genetic predisposition to DR [9,10]. However, to date, genome-wide association studies (GWAS) have yielded results that are not reproducible in replicates or other populations. Among the reasons for the ambiguity of the obtained data are the insufficient sample size, differences in the classification of patients, and ethnic features. Nevertheless, it is clear from the results that the genetic architecture of this disease is very complex and is associated with numerous environmental risk factors and interactions between genes and the environment [10,11].

Previous studies have identified associations of several polymorphisms with a predisposition to microvascular complications of DM in various world populations [12,13,14,15,16,17,18,19,20]. In this research, we tested the hypothesis of an association of some of these polymorphisms with the risk of DR development in the Kazakhstan population. For this, in accordance with the available resources, we selected five polymorphisms (−2549 I/D in the VEGF gene, −429 T/C and −374 T/A in the RAGE gene, rs7903146 in the TCF7L2 gene, BglII in the ITGA2 gene) and conducted a case–control study comparing allele and genotype frequencies in three groups: DR patients, T2DM patients without DR, and healthy controls.

## 2. Materials and Methods

### 2.1. Study Subjects

Venous blood sampling of 94 T2DM patients with clinically confirmed DR and 94 T2DM patients without DR was carried out in three medical institutions in Almaty in 2020–2021: Almaty Multidisciplinary Clinical Hospital, City Center of Diabetes and Kazakh Research Institute of Eye Diseases of Ministry of Healthcare of the Republic of Kazakhstan. The exclusion criteria were: type 1 DM, oncological, severe cardiovascular, respiratory, renal, infectious, and mental diseases, pregnancy and lactation, and childhood.

After taking the anamnesis and a physical examination, all of the patients underwent a complete ophthalmological examination, including visometry, refractometry, keratometry, tonometry, perimetry, biomicroscopy, ophthalmoscopy, and optical coherence tomography. Retina examination was performed by indirect ophthalmoscopy under mydriasis with an aspherical lens. The examination results were classified into four stages according to the classification of Kohner E. and Porta M.: no retinopathy, non-proliferative DR (NPDR), pre-proliferative DR (PPDR), and proliferative DR (PDR). Blood sampling of 51 healthy Kazakh people (controls) was carried out at the Karasay Central District Hospital in Almaty Region in 2019. 

This study complies with the principles stated in the Declaration of Helsinki and was approved by the local ethics committee of M. Aitkhozhin Institute of Molecular Biology and Biochemistry. All of the participants signed informed consent for the use of the biomaterial in this study.

### 2.2. SNPs Selection

Given the available resources, the following criteria were used when selecting SNPs: (1) the SNP is associated with microvascular complications of T2DM according to previous studies; (2) it is feasible to use the PCR-RFLP method for genotyping; (3) the minor allele frequency of the SNP is high (MAF > 0.2) to allow for a relatively small sample size. As a result, five polymorphisms were selected for testing: −2549 I/D in the VEGF gene, −429 T/C and −374 T/A in the RAGE gene, rs7903146 in the TCF7L2 gene, BglII in the ITGA2 gene.

### 2.3. DNA Isolation and Genotyping 

The DNA was isolated from the blood using a commercial DNeasy Blood & Tissue Kit manufactured by QIAGEN (Hilden, Germany) according to the manufacturer’s protocol. Genotyping was performed using the PCR-RFLP (restriction fragment length polymorphism) method. The sequences of primers for the amplification of the necessary fragments were taken from earlier works [12,21,22,23]. The amplification products were treated with the appropriate restriction endonuclease. The fragment lengths were evaluated by electrophoresis in 8% polyacrylamide gel.

### 2.4. Statistics

To compare the characteristics of the studied groups, we used Student’s *t*-test (for quantitative data) and Pearson’s goodness-of-fit test (χ^2^-test) (for nominal data). The correspondence of the genotype distribution to the Hardy–Weinberg law and the significance of differences in the frequencies of genotypes and alleles among groups were calculated using the χ^2^-test. The odds ratio (OR) with a 95% confidence interval was used as an indicator of the degree of association. Logistic regression analysis was performed to assess the independent role of genotypes as a risk factor. *p* < 0.05 was considered statistically significant. For multiple comparisons, the Bonferonni method was used to adjust the *p*-values. The Jamovi v2.2.5.0 software was used for the calculations [24]. Statistical power was estimated using the Epi Info™ v7.2.5.0 software [25].

## 3. Results

In this study, we tested the hypothesis about an association of five polymorphisms of the VEGF (-2549 I/D), RAGE (-429 T/C and -374 T/A), TCF7L2 (rs7903146), and ITGA2 (BglII) genes with a predisposition to DR among T2DM patients in the Kazakhstan population. The allele and genotype frequencies were compared between groups of DR patients, T2DM patients without DR, and healthy controls.

### 3.1. Characteristics of the Compared Groups

The characteristics of the compared groups are presented in Table 1. Comparing the demographic and clinical characteristics between the groups of DR patients and T2DM patients without DR, we found no statistically significant differences in almost all indicators (data not shown), except for the duration of diabetes: at the time of the study, the group of DR patients had on average of 4 years of illness longer than the group of T2DM patients without complications in the fundus (*p* = 1.47 × 10^−4^). A comparison of groups of T2DM patients (with DR and without DR) with the group of healthy controls revealed differences in age (*p* = 3.19 × 10^−7^ and 6.82 × 10^−7^, respectively). Additionally, DR patients and controls differed significantly in sex distribution (*p* = 0.016). It should be noted that when comparing nominal data, we had the statistical power to detect only large differences (13–20% and above).

### 3.2. Genotype Distribution among Groups

Using the PCR-RFLP method, we determined the genotypes of all individuals from three groups (Appendix A). The genotype distributions of the studied polymorphisms are presented in Table 2. The distributions in each case corresponded to the Hardy–Weinberg equilibrium.

There were no statistically significant differences in the genotype distributions of all studied polymorphisms between the groups of DR patients and T2DM patients without DR. Likewise, there were no significant differences in the genotype distributions between groups of DR patients with and without DME, with different degrees of DR progression and duration of diabetes (data not shown). According to our data, none of the five polymorphisms are associated with the development of DR in T2DM patients in the Kazakhstani population.

Next, we compared both groups of T2DM patients (with and without DR) with healthy controls and found no significant differences in genotype distributions for four polymorphisms in the VEGF, RAGE, and TCF7L2 genes. Otherwise, there were significant differences in genotype distributions for the ITGA2 gene BglII polymorphism in both comparisons (*p* = 0.042 and 0.005, respectively), although after adjusting for multiple comparisons, only differences between T2DM patients without DR and healthy controls remained significant (*p_Bonf_* = 0.027).

### 3.3. Association Analysis of the ITGA2 Gene BglII Polymorphism 

The revealed differences in the genotype distribution of the ITGA2 gene BglII polymorphism between the groups of T2DM patients (with and without DR) and healthy controls were also observed when comparing the allele frequencies (*p* = 0.021 and 0.002, respectively). Table 3 shows the results of the association analysis of this polymorphism. We did not find an association between the BglII polymorphism and DR. However, according to the OR values, BglII polymorphism is associated with T2DM, and the major allele BglII− should be considered as the risk allele (OR for comparison of DR patients and controls = 1.81 [95% CI: 1.09–2.99], *p* = 0.029 and OR for comparison of T2DM patients without DR and controls = 2.25 [95% CI: 1.35–3.77], *p* = 0.003). The frequency of the BglII−/− genotype in both groups of T2DM patients (with and without DR) was significantly higher than in the healthy controls (53.2 and 55.3%, respectively, vs. 31.4%, OR = 2.49 [1.21–5.09], *p* = 0.019 and 2.71 [1.32–5.55], *p* = 0.010). The most appropriate inheritance model was the recessive model.

To eliminate the factor of ethnic heterogeneity, we conducted the same comparisons in a subset of Kazakhs. Comparisons were not made for other ethnic groups due to the small representation. Studying in groups of Kazakhs showed similar results as in the general sample (Table 4). Among Kazakhs, there were no differences in the genotype distributions between the groups of DR patients and T2DM patients without DR. Comparing both groups of diabetic patients (with and without DR) with healthy controls revealed statistically significant differences in the genotype distributions (*p* = 0.022 and 0.043, respectively) and allele frequencies (*p* = 0.013 and 0.018, respectively), although in three out of four cases the differences were less significant than in the general sample. The revealed association of the BglII− allele and BglII−/− genotype with T2DM have also been observed in the subset of Kazakhs, and the OR values were similar to those obtained for the total mixed sample.

Table 5 shows the OR values adjusted for the three available parameters using logistic regression analysis. According to the data, BglII (−/−) genotypes are independent risk factors for type 2 diabetes mellitus. Despite highly significant *p*-values, age cannot be considered a risk factor. As noted above, the control group does not correspond in age to the groups of T2DM patients (with and without DR). Apparently, for this reason, age in the logistic regression model appears as a risk factor.

Our results indicate an association of the ITGA2 gene BglII polymorphism with the development of T2DM. It is likely that the sample size was insufficient to identify the association of this polymorphism with DR. This is supported by an apparent excess of the homozygous BglII+/+ genotype in the group of DR patients compared to T2DM patients without DR: 11% vs. 4%, OR = 2.60 [95% CI: 0.77–8.83], *p* = 0.198.

## 4. Discussion

Hyperglycemia in DM causes the dysregulation of several signaling pathways that affect the functioning of the receptors of the cells of the blood coagulation system [25,26]. The platelets of diabetic patients are hyperreactive to activating agents such as adenosine diphosphate, collagen, and thrombin. Increased thrombus formation observed in DM patients is one of the main factors in the pathogenesis and progression of vascular complications, including microvascular ones [26,27,28]. Vessel damage leads to exposure of the collagen-rich subendothelial layer, which interacts with platelet receptors and causes their direct adhesion to the damaged subendothelium and activation [29]. Considering the important role of integrins, transmembrane heterodimeric cellular receptors for extracellular matrix ligands, in the activation of thrombus formation, it might be assumed that polymorphisms in integrin genes would be associated with DR. Indeed, several independent studies in different populations have shown that polymorphisms in the ITGA2 gene of the α-subunit of the α2β1 integrin, collagen, and laminin receptor, are associated with the risk of DR [30,31,32,33].

According to Kunicki et al. [34], the amount of the α2β1 integrin receptor on the surface of platelets can vary up to 10-fold in different people and is associated with the silent transition of T807C in the ITGA2 gene of the integrin α-subunit: the T allele is associated with an increase, while the C allele is associated with the reduced density of the receptor. Later, in another report, the same research team showed that this relationship is probably caused by linkage disequilibrium with another C-52T substitution in the 5’-regulatory region of the gene, which leads to a decrease in its expression by affecting the regulatory proteins Sp1 and Sp3 [35]. In addition to these two substitutions, at least six polymorphisms are known directly in the ITGA2 gene and its regulatory regions, which are also in linkage disequilibrium with each other. Among them, the G/A transition (rs2910964) is located in intron 7, more known as the BglII polymorphism. According to the literature data, the major BglII− allele is linked to the 807C and −52T alleles and is associated with a reduced receptor density, while the minor BglII+ allele is linked to the 807T and −52C alleles and is associated with an increased receptor density [29,30].

Our study did not reveal a relationship between the BglII polymorphism and DR in the Kazakhstan population. These data are inconsistent with some of the earlier studies. Matsubara et al. identified an association of the BglII+ allele with the development of DR among the Japanese, and the association increased in a subset of patients with ≥10 years of T2DM [30]. Petrovič et al., on a sample of Europeans, showed that the BglII+/+ genotype could be considered an independent risk factor for the development of DR [30]. The same conclusion was reached by Midani et al., who conducted the study on a sample of Tunisians [32], and Azmy et al. on a mixed sample of Egyptians [33].

On the other hand, there is evidence in the literature supporting our data. Li et al. in the Chinese population did not find significant differences in the genotype distributions of BglII polymorphism between the DR patients and the T2DM patients with disease duration of more than 10 years but without complications in the fundus [36]. Iskhakova et al. found no association of BglII polymorphism with DR among T2DM patients in the Volga region population, most (84%) of which were Russians [37].

In our study, we showed that BglII polymorphism could be associated not only with DR but also directly with T2DM. In addition, according to our data, the risk allele for T2DM is the wild BglII− allele and not the minor BglII+ allele, which is considered risky in conditions with pathologically hyperactive thrombus formation, as in the case of DR, as well as other vascular diseases, such as stroke [38]. The allele frequencies of this SNP in the Kazakh ethnic group (MAF 0.42) obtained in this study are closer to those for European populations (MAF 0.41 according to NCBI) than for East Asian populations (MAF 0.28, NCBI). In the Kazakh population, patients with type 2 diabetes showed a significant shift in the allelic ratio to MAF 0.26–0.27.

In all the above works, the authors did not compare the frequency of the ITGA2 gene BglII polymorphism between groups of T2DM patients and healthy individuals. In the only earlier work, Afzal et al., studying the Pakistani population, in addition to T2DM patients with and without retinopathy, also included in the analysis a group of healthy controls. As a result, significant differences in the genotype distributions between DR patients and controls were revealed; however, in contrast to our study, the frequency of the minor allele BglII+ in DR patients was increased, and this allele was considered risky for DR [39].

Obviously, the sample size was a limitation of the study. Considering the reduced frequencies of the BglII+ allele in Kazakh T2DM patients, it is likely that the limited sample size did not allow us to detect a statistically sufficient number of BglII+/+ homozygotes to identify an association with DR. It should be noted that there is a tendency for the prevalence of the BglII+/+ genotype in DR patients compared to T2DM patients without retinopathy. Therefore, further studies with larger sample sizes are needed.

The other four studied polymorphisms also did not show the expected association with DR in the Kazakh population, which is not consistent with the results of previous studies in other world populations. The inconsistency of our data with these works confirms the thesis about the complexity of the genetic architecture of T2DM and its microvascular complications.

## 5. Conclusions

Overall, ITGA2 gene polymorphism plays an important role in the pathogenesis of DR, as confirmed by the data of several independent studies. This study has shown that the ITGA2 gene BglII polymorphism can be associated not only with DR but also with T2DM. According to our data, the risk allele for T2DM is the wild BglII− allele, and not the minor BglII+, which is considered risky for DR.

## Figures and Tables

**Table 1 medicina-58-01416-t001:** Demographic and clinical characteristics of the study groups.

Characteristics	T2DM Patients with DR (A)	T2DM Patients without DR (B)	Healthy Controls (C)	A vs. B *p*-Value	A vs. C *p*-Value	B vs. C *p*-Value
Total (No.)	94	94	51			
of them:						
Female sex (%)	63.8	57.4	43.1	0.370	0.016	0.099
Kazakhs (%)	62.8	68.1	100	-	-	-
Uighurs (%)	16.0	13.8	-	-	-	-
Russians (%)	14.9	10.6	-	-	-	-
Other ethnic groups * (%)	6.4	7.4	-	-	-	-
Smokers (%)	8.5	18.1	-	0.053	-	-
Drinkers (on holidays) (%)	13.8	13.8	-	1	-	-
With a family history of diabetes (%)	46.8	45.7	-	0.884	-	-
NPDR ^1^ (%)	38.3	-	-	-	-	-
PPDR ^2^ (%)	40.4	-	-	-	-	-
PDR ^3^ (%)	21.3	-	-	-	-	-
DME ^4^ (%)	19.1	-	-	-	-	-
Age	60.36 ± 10.59	59.53 ± 9.75	51.75 ± 5.94	0.577	3.19 × 10^−7^	6.82 × 10^−7^
Age of onset	47.84 ± 12.11	50.85 ± 10.75	-	0.073	-	-
Disease duration	12.51 ± 6.52	8.60 ± 7.29	-	1.47 × 10^−4^	-	-
Body mass index, female	28.27 ± 4.71	28.84 ± 6.03	-	0.572	-	-
Body mass index, male	28.02 ± 3.49	26.96 ± 4.78	-	0.287	-	-
Blood glucose level (fasting), mmol/L	9.82 ± 3.70	9.77 ± 3.20	-	0.914	-	-

* Koreans, Kurds, Azerbaijanis, Uzbeks, Karakalpaks, Chinese, Turks, Udmurts. ^1^ non-proliferative DR; ^2^ pre-proliferative DR; ^3^ proliferative DR; ^4^ diabetic macular edema.

**Table 2 medicina-58-01416-t002:** Genotype distribution of the studied polymorphisms.

Polymorphism	Genotype	DR (*n* = 94) *n* (%)	T2DM (*n* = 94) *n* (%)	Healthy (*n* = 51) *n* (%)	DR vs. T2DM χ^2^/*p*-Value	DR vs. Healthy χ^2^/*p*-Value	T2DM vs. Healthy χ^2^/*p*-Value
−2549 I/D in VEGF	DD	34 (0.36)	39 (0.41)	22 (0.43)	0.563/0.755	0.743/0.690	0.081/0.960
	DI	45 (0.48)	41 (0.44)	21 (0.41)			
	II	15 (0.16)	14 (0.15)	8 (0.16)			
	tHWE ^1^, *p*-value	1.000	0.911	0.744			
−374 T/A in RAGE	AA	45 (0.48)	53 (0.56)	33 (0.65)	1.453/0.484	3.779/0.151	0.962/0.618
	AT	44 (0.47)	36 (0.38)	16 (0.31)			
	TT	5 (0.05)	5 (0.05)	2 (0.04)			
	tHWE ^1^, *p*-value	0.574	0.967	0.999			
−429 T/C in RAGE	AA	75 (0.80)	76 (0.81)	39 (0.76)	1.007/0.605	2.869/0.238	5.650/0.059
	AG	18 (0.19)	18 (0.19)	9 (0.18)			
	GG	1 (0.01)	0 (0.00)	3 (0.06)			
	tHWE ^1^, *p*-value	0.999	0.602	0.211			
rs7903146 in TCF7L2	CC	55 (0.59)	52 (0.55)	36 (0.71)	0.251/0.882	3.525/0.172	4.896/0.086
	CT	35 (0.37)	37 (0.39)	15 (0.29)			
	TT	4 (0.04)	5 (0.05)	0 (0.00)			
	tHWE ^1^, *p*-value	0.923	0.938	0.281			
BglII in ITGA2	(−/−)	50 (0.53)	52 (0.55)	16 (0.31)	2.833/0.243	6.347/0.042	10.418/0.005
	(−/+)	34 (0.36)	38 (0.40)	27 (0.53)		0.209 *	0.027 *
	(+/+)	10 (0.11)	4 (0.04)	8 (0.16)			
	tHWE ^1^, *p*-value	0.722	0.789	0.832			

^1^ Test for Hardy–Weinberg equilibrium; * *p* after Bonferroni adjustment for multiple testing.

**Table 3 medicina-58-01416-t003:** Association analysis of the BglII polymorphism in ITGA2 gene.

		DR *n* (%)	T2DM *n* (%)	Healthy *n*s (%)	DR vs. T2DM		DR vs. Healthy		T2DM vs. Healthy	
					OR (95%CI)	*p*-Value	OR (95%CI)	*p*-Value	OR (95%CI)	*p*-Value
Additive model	(−/−)	50 (53.2)	52 (55.3)	16 (31.4)	1	-	1	-	1	-
	(−/+)	34 (36.2)	38 (40.4)	27 (52.9)	0.93 (0.51–1.70)	0.937	0.40 (0.19–0.86)	0.028	0.43 (0.21–0.91)	0.042
	(+/+)	10 (10.6)	4 (4.3)	8 (15.7)	2.60 (0.77–8.83)	0.198	0.40 (0.14–1.19)	0.165	0.15 (0.04–0.58)	0.008
Recessive model	(−/−)	50 (53.2)	52 (55.3)	16 (31.4)	0.92 (0.52–1.63)	0.884	2.49 (1.21–5.09)	0.019	2.71 (1.32–5.55)	0.010
	(−/+) + (+/+)	44 (46.8)	42 (44.7)	35 (68.6)	1	-	1	-	1	-
Alleles	(−)	134 (71.3)	142 (75.5)	59 (57.8)	0.80 (0.51–1.27)	0.414	1.81 (1.09–2.99)	0.029	2.25 (1.35–3.77)	0.003
	(+)	54 (28.7)	46 (24.5)	43 (42.2)	1	-	1	-	1	-

**Table 4 medicina-58-01416-t004:** Association analysis of the ITGA2 gene BglII polymorphism in Kazakhs group.

		DR *n* (%)	T2DM *n* (%)	Healthy *n* (%)	DR vs. T2DM		DR vs. Healthy		T2DM vs. Healthy	
					OR (95% CI)	*p*-Value	OR (95% CI)	*p*-Value	OR (95% CI)	*p*-Value
Additive model	(−/−)	34 (57.6)	32 (50.0)	16 (31.4)	1	-	1		1	
	(−/+)	19 (32.2)	29 (45.3)	27 (52.9)	0.62 (0.29–1.31)	0.284	0.33 (0.14–0.76)	0.015	0.54 (0.24–1.19)	0.181
	(+/+)	6 (10.2)	3 (4.7)	8 (15.7)	1.88 (0.43–8.17)	0.618	0.35 (0.11–1.19)	0.160	0.19 (0.04–0.80)	0.040
Recessive model	(−/−)	34 (57.6)	32 (50.0)	16 (31.4)	1.36 (0.67–2.77)	0.505	2.98 (1.36–6.52)	0.010	2.19 (1.02–4.72)	0.068
	(−/+) + (+/+)	25 (42.4)	32 (50.0)	35 (68.6)	1	-	1	-	1	-
Alleles	(−)	87 (73.7)	93 (72.6)	59 (57.8)	1.06 (0.60–1.86)	0.964	2.05 (1.16–3.61)	0.019	1.94 (1.11–3.37)	0.027
	(+)	31 (26.3)	35 (27.3)	43 (42.2)	1	-	1	-	1	-

**Table 5 medicina-58-01416-t005:** Odds ratios (OR) adjusted by logistic regression analysis for the association with type II diabetes mellitus.

Risk Factors	DR vs. Healthy		T2DM vs. Healthy	
	OR (95% CI)	*p*-Value	OR (95% CI)	*p*-Value
Age	1.11 (1.06–1.17)	5.11 × 10^−5^	1.11 (1.06–1.17)	3.04 × 10^−5^
BglII (−/−) genotype	2.31 (1.05–5.08)	0.038	2.66 (1.20–5.88)	0.016
Male sex	0.63 (0.29–1.36)	0.236	0.86 (0.39–1.88)	0.859

## Data Availability

Data are contained within the article and Appendix A.

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
