# Peer review of "ITGA2 Gene Polymorphism Is Associated with Type 2 Diabetes Mellitus in the Kazakhstan Population"

_medicina, 2022, doi:10.3390/medicina58101416_

Round 1

Reviewer 1 Report

In this manuscript, Magazova and colleagues investigated the association between polymorphisms in the VEGF, RAGE, TCF7L2 and ITGA2 genes and type 2 diabetes and diabetic retinopathy in a Kazakhstani population. Their studies were conducted on 94 diabetic patients with retinopathy (DR) and 94 without DR and 51 healthy controls. None of the polymorphisms tested was found to differ from HW. The DR-affected and DR-free groups did not differ significantly. Of the 5 polymorphisms tested, only the one in the ITGA2 gene showed a significant association with the presence of T 2DM. The result remained significant after test correction regardless of the presence or absence of DR.

Comments and suggestions:

1.       Throughout the manuscript, "p" is the lower case for the p-value.

2.     Table 1 should be in the Results section instead of the Materials and Methods section.

3.       For Table 1, it would be useful to indicate the % instead of the sample number. As well as a statistical test to show possible differences.

4.       Please indicate the meaning of the abbreviations in Table 1 as footnotes.

5.       How were the polymorphisms selected? What was the reason for testing these and not others?

6.       Logistic regression requires the data to be normally distributed. Has this been tested, was any deviation from the normal distribution found, and was the data transformed?

7.       For the results presented in Table 3, were adjustments made for age, sex, BMI, and age of onset of DM?

8.       Although only the ITGA2 gene polymorphism showed a significant difference based on the genotype distribution, it cannot be concluded that the other polymorphisms studied are not associated with the development of T2DM or DM. Was the logistic regression analysis for the ITGA2 gene polymorphism performed for the others?

9.       The effects of the polymorphisms tested are generally small in themselves. Perhaps a combined analysis of the 5 polymorphisms using a genetic risk score or an optimised genetic risk score would provide a stronger association.

Overall, the results presented in this manuscript are interesting and raise questions despite the relatively low sample sizes. The low number of polymorphisms tested is a limitation that needs to be compensated by complex analysis.

Reviewer 2 Report

This paper performed a case-control study in Kazakhstan population (consisting of 94 diabetic patients with DR, 94 diabetic patients without DR, and 51 healthy controls.) to evaluate the association of VEGF, RAGE, TCF7L2 and ITGA2 polymorphisms with diabetes. They demonstrated that the BglII polymorphism in ITGA2 gene can be associated with T2DM, but not with DR in the Kazakhstan population.

This paper is interesting. However, some questions are confusing.

1.       Although many papers have been published concerning this issue, the role of VEGF (-2549I/D), RAGE (-429T/C and -374T/A), TCF7L2 (rs7903146), and ITGA2 (BglII) genes polymorphisms in diabetes remains controversial. Thus, to avoid making more confusions, a case-control study performed in a large cohort is preferred. Obviously, the sample size of this study is small and findings from this study will only lead to more inconclusive results.

2.       Statistical analysis. Please give here the estimates you used for the power calculation. you need to calculate whether you have enough statistical power with your sample size to detect any significant difference in the clinical parameters you have analyzed.

3.       PCR-RFLP genotyping results should be presented in the manuscript.

Round 2

Reviewer 1 Report

I accept the Authors' answers to my questions 1 to 6.

My further comments:

- For Table 5, what justifies the correction for age and sex but not for BMI, which is considered a significant risk factor for T2DM? Why is the correction for BMI not included in the analysis presented in Table 5?

- In response to my 8. comment, I would have liked to have seen the results of the analyses, whether or not they were included in the final manuscript.

- To calculate the genetic risk score, it is not necessary to verify the effect of the SNP by GWAS analysis. In fact, knowledge of effect sizes is necessary to calculate weighted GRS, but knowledge of risk alleles is sufficient to calculate unweighted GRS. There are no practical problems in performing an unweighted GRS calculation.

Round 3

Reviewer 1 Report

I accept the answers given by the Authors to my questions and comments.